# Antitumor Activity by an Anti-CD44 Variant 9 Monoclonal Antibody in Gastric and Colorectal Cancer Xenograft Models

**DOI:** 10.3390/ijms26189170

**Published:** 2025-09-19

**Authors:** Mayuki Tawara, Hiroyuki Suzuki, Tomokazu Ohishi, Mika K. Kaneko, Yukinari Kato

**Affiliations:** 1Department of Antibody Drug Development, Tohoku University Graduate School of Medicine, 2-1 Seiryo-machi, Aoba-ku, Sendai 980-8575, Japan; tawara.mayuki.p8@dc.tohoku.ac.jp (M.T.); mika.kaneko.d4@tohoku.ac.jp (M.K.K.); 2Institute of Microbial Chemistry (BIKAKEN), Laboratory of Oncology, Microbial Chemistry Research Foundation, 3-14-23 Kamiosaki, Shinagawa-ku, Tokyo 141-0021, Japan; ohishit@bikaken.or.jp

**Keywords:** CD44v9, monoclonal antibody therapy, ADCC, CDC, gastric cancer, colorectal cancer

## Abstract

CD44 variants (CD44v) play essential roles in the promotion of tumor metastasis, maintenance of cancer stem cell properties, and resistance to treatments. Therefore, the development of anti-CD44v mAbs is essential for targeting CD44v-positive tumor cells. An anti-CD44v9 mAb, C_44_Mab-1 (mouse, IgG_1_, kappa), was previously established. C_44_Mab-1 recognizes the variant exon 9-encoded region and applies to multiple research techniques. A mouse IgG_2a_ version of C_44_Mab-1 (C_44_Mab-1-mG_2a_) was generated to evaluate the in vitro and in vivo antitumor activities using gastric and colorectal cancer cell lines. C_44_Mab-1-mG_2a_ showed a reactivity to CD44v3–10-overexpressed Chinese hamster ovary-K1 (CHO/CD44v3–10), gastric cancer MKN45, and colorectal cancer COLO205 in flow cytometry. C_44_Mab-1-mG_2a_ exhibited both antibody-dependent cellular cytotoxicity (ADCC) and complement-dependent cytotoxicity (CDC) against CHO/CD44v3–10, MKN45, and COLO205. Furthermore, administration of C_44_Mab-1-mG_2a_ significantly suppressed CHO/CD44v3–10, MKN45, and COLO205 xenograft tumor growth compared with control mouse IgG_2a_. These results indicated that C_44_Mab-1-mG_2a_, which possesses ADCC/CDC activities, could be applied to the mAb-based therapy against CD44v9-positive carcinomas.

## 1. Introduction

A type I transmembrane glycoprotein, CD44, was initially characterized in the 1980s as a lymphocyte homing receptor [1]. The human CD44 gene is located on chromosome 11p13. The exons 1–5 encode the conserved extracellular domain, exons 16 and 17 encode the stalk region, exon 18 encodes the transmembrane domain, and exons 19 and 20 encode the intracellular cytoplasmic domain. The isoform encoded by exons 1–5 and 16–20 is referred to as the standard form (CD44s), which is present in most types of cells. Exons 6–15 are subject to alternative splicing, generating multiple variant (CD44v) isoforms by insertion between the extracellular and stalk regions [2]. CD44 is broadly expressed in lymphocytes, fibroblasts, and smooth muscle cells. In normal epithelial tissues, various CD44v isoforms are detected, with variant 9 (v9) being the most prevalent, followed by v6 and v4 [3]. Aberrant expression of CD44v has been implicated in tumor progression [4]. CD44v8-10 (CD44E) is predominantly expressed in epithelial cells, whereas CD44v3–10, the largest isoform, is mainly found in keratinocytes [5].

Both CD44s and CD44v (pan-CD44) bind to hyaluronic acid (HA) through the conserved extracellular domain, which plays critical roles in cellular homing, migration, adhesion, and proliferation [6]. The variant isoforms of CD44 (CD44v) have been implicated in multiple oncogenic processes, including the promotion of tumor invasion and metastasis [7], acquisition of cancer stem cell (CSC) properties [8], and resistance to chemotherapy and radiotherapy [9]. The v3-encoded region undergoes heparan sulfate modification, enabling high-affinity binding to heparin-binding growth factors, such as fibroblast growth factors. This modification allows the v3 region to function as a co-receptor for receptor tyrosine kinases, thereby enhancing downstream signal transduction [10]. The v6-encoded region is essential for c-MET activation through the formation of a ternary complex with hepatocyte growth factor [11]. In addition, the v8–10-encoded region interacts with and stabilizes the cystine–glutamate transporter (xCT), facilitating cystine uptake and subsequent glutathione synthesis, which mitigates reactive oxygen species (ROS)-induced stress [12]. Regulation of redox homeostasis through CD44v8–10–xCT interaction is associated with poor clinical prognosis [13]. Accordingly, the generation and characterization of monoclonal antibodies (mAbs) that recognize each variant exon are essential for elucidating their distinct biological functions and for the development of CD44-targeted cancer therapies.

CD44 is also recognized as a cell surface marker of cancer stem-like cells (CSCs) in various carcinomas [5]. CD44s or CD44v isoform-specific mAbs have been employed to isolate CD44-high CSC populations [5]. These CD44-high cells exhibit CSC properties, including drug resistance and robust tumorigenic potential in vivo [5]. Consequently, the development of anti-CD44 mAbs capable of selectively recognizing individual variants is crucial for elucidating CSC biology and developing variant-specific cancer therapy.

Multiple isoforms of CD44 have been implicated in malignant progression across diverse tumor types [7], including colorectal cancers [14], pancreatic cancers [15,16], prostate cancers [17], head and neck squamous cell carcinomas [18], breast cancers [19], and gliomas [20,21]. Furthermore, a comprehensive multi-omic analysis of malignant gastric cancers revealed genomic amplifications of established cancer driver genes such as EGFR, ERBB2, MET, FGFR2, and CD44 in gastric cancer with peritoneal metastasis [10]. These cell surface antigens are potential therapeutic targets for mAb-based interventions [11]. While mAb therapies and diagnostic approaches have been developed for the first four antigens, equivalent strategies targeting CD44 remain largely unestablished.

We previously generated highly sensitive and specific mAbs targeting CD44 by immunizing mice with stably CD44v3–10-overexpressed Chinese hamster ovary-K1 (CHO/CD44v3–10). The critical epitopes recognized by these mAbs were identified through enzyme-linked immunosorbent assay, followed by functional characterization in flow cytometry, western blotting, and immunohistochemistry. Among the established clones, C_44_Mab-1 (mouse IgG_1_, κ) bound to a peptide corresponding to the v9-encoded region. Flow cytometric analysis demonstrated that C_44_Mab-1 recognized CHO/CD44v3–10 cells as well as colorectal cancer cell lines (COLO201 and COLO205). Furthermore, C_44_Mab-1 successfully detected endogenous CD44v9 in colorectal cancer tissues by immunohistochemistry [22]. These findings indicate that C_44_Mab-1 is a valuable tool for detecting CD44v9 across multiple applications.

In this study, we converted C_44_Mab-1 into a mouse IgG_2a_ type mAb (C_44_Mab-1-mG_2a_). We evaluated antibody-dependent cellular cytotoxicity (ADCC), complement-dependent cytotoxicity (CDC), and antitumor effect in gastric and colorectal cancer xenograft models.

## 2. Results

### 2.1. Production of Mouse IgG_2a_-Type Anti-CD44v9 mAb, C_44_Mab-1-mG_2a_

Previously we established an anti-CD44v9 mAb, C_44_Mab-1, by immunizing mice with CHO/CD44v3–10. C_44_Mab-1 recognizes the CD44v variant exon 9-encoded region and shows a high binding affinity against CHO/CD44v3–10 and tumor cells [22]. In this study, the complementarity-determining regions (CDRs) of C_44_Mab-1 were first determined from the cDNA of C_44_Mab-1-producing hybridoma. To evaluate the antitumor activity, a mouse IgG_2a_-type C_44_Mab-1 (C_44_Mab-1-mG_2a_) was constructed by fusing the V_H_ and V_L_ CDRs of C_44_Mab-1 with the C_H_ and C_L_ chains of mouse IgG_2a_ (Figure 1A). As a control mouse IgG_2a_ (mIgG_2a_), PMab-231 (an anti-tiger podoplanin mAb, mouse IgG_2a_) was produced as described previously [23]. We confirmed the purity of the recombinant mAbs by SDS-PAGE under reduced conditions (Figure 1B). Figure 1C illustrates the recognition regions of the mAbs (C_44_Mab-1-mG_2a_, 5-mG_2a_, and C_44_Mab-46-mG_2a_) used in this study.

### 2.2. Flow Cytometry Using C_44_Mab-1-mG_2a_

The reactivity of C_44_Mab-1-mG_2a_ was first confirmed using CHO/CD44v3–10. As shown in Figure 2A, C_44_Mab-1-mG_2a_ showed dose-dependent reactivity to CHO/CD44v3–10, but not to CHO/CD44s or CHO-K1. We confirmed the expression of CD44s by a pan-CD44 mAb, 5-mG_2a_ (Appendix A). Control mIgG_2a_ did not recognize CHO/CD44v3–10 (Figure 2B). Next, the binding affinity was investigated. The dissociation constant (*K*_D_) value of C_44_Mab-1-mG_2a_ for CHO/CD44v3–10 was determined to be 4.7 × 10^−9^ M (Figure 2C). These results indicated that C_44_Mab-1-mG_2a_ possesses comparable reactivity and affinity with parental mAb, C_44_Mab-1 as reported previously (*K*_D_: 2.5 × 10^−8^ M [22]).

C_44_Mab-1 could recognize colorectal cancer cell lines, including COLO205, in flow cytometry [22]. In addition to colorectal cancer, the reactivity of C_44_Mab-1-mG_2a_ to gastric cancer MKN45 was investigated in flow cytometry. As shown in Figure 3A, C_44_Mab-1-mG_2a_ exhibited comparable reactivity to MKN45 and COLO205. Control mIgG_2a_ did not recognize MKN45 and COLO205 (Figure 3B). C_44_Mab-1-mG_2a_ also reacted with other gastric cancer cell lines (LMSU, KatoIII, and NUGC-4, Appendix A). These results indicated that C_44_Mab-1-mG_2a_ retains the reactivity to CD44v9-positive cells.

### 2.3. Immunohistochemistry of Gastric Cancer Using C_44_Mab-1-mG_2a_

We previously stained a colorectal cancer tissue array using C_44_Mab-1 [22]. Therefore, we next investigated the CD44v9 expression in the gastric cancer tissue array (BS01011b) using C_44_Mab-1-mG_2a_ and an anti-pan CD44 mAb, C_44_Mab-46-mG_2a_ [24]. C_44_Mab-1-mG_2a_ exhibited membranous staining in intestinal-type gastric cancer (Figure 4A, left). C_44_Mab-46-mG_2a_ stained the same type of cancer cells and surrounding stroma cells (Figure 4A, right). In diffuse-type gastric cancer, diffusely spread tumor cells were strongly stained by both C_44_Mab-1-mG_2a_ and C_44_Mab-46-mG_2a_ (Figure 4B). Additionally, stromal staining by C_44_Mab-46 was also observed in the tissue (Figure 4B, right).

We summarized the immunohistochemistry of gastric cancer tissue array in Appendix A; C_44_Mab-1-mG_2a_ stained 52 out of 72 cases (72%) of gastric cancer but did not stain the stromal tissues. These results indicated that C_44_Mab-1-mG_2a_ is also helpful to detect CD44v9 in immunohistochemistry of formalin-fixed paraffin-embedded gastric cancers.

### 2.4. ADCC and CDC by C_44_Mab-1-mG_2a_

The ADCC caused by C_44_Mab-1-mG_2a_ against CHO/CD44v3–10, MKN45, and COLO205 was investigated. The splenocytes derived from BALB/c nude mice were used as an effector. C_44_Mab-1-mG_2a_ showed potent ADCC against CHO/CD44v3–10 (37.2% vs. 9.4% cytotoxicity of control mIgG_2a_, *p* < 0.05, Figure 5A), MKN45 (33.1% vs. 14.6% cytotoxicity of control mIgG_2a_, *p* < 0.05, Figure 5B), and COLO205 (15.0% vs. 3.9% cytotoxicity of control mIgG_2a_, *p* < 0.05, Figure 5C). We also confirmed the effector cell activation by an ADCC reporter bioassay. Appendix A showed that the effector cells were activated by C_44_Mab-1-mG_2a_ in the presence of CHO/CD44v3–10, but not CHO-K1. The effector cells were not activated by control mIgG_2a_.

The CDC caused by C_44_Mab-1-mG_2a_ against CHO/CD44v3–10, MKN45, and COLO205 was next examined. C_44_Mab-1-mG_2a_ showed significant CDC against CHO/CD44v3–10 (51.6% vs. 12.1% cytotoxicity of control mIgG_2a_, *p* < 0.05, Figure 6A), MKN45 (13.3% vs. 3.4% cytotoxicity of control mIgG_2a_, *p* < 0.05, Figure 6B), and COLO205 (22.5% vs. 13.0% cytotoxicity of control mIgG_2a_, *p* < 0.05, Figure 6C. These results indicated that C_44_Mab-1-mG_2a_ exerts antitumor efficacy through activation of effector cells and complements in vitro.

### 2.5. Antitumor Effect by C_44_Mab-1-mG_2a_ Against CHO/CD44v3–10, MKN45, and COLO205 Xenografts

The antitumor effects caused by C_44_Mab-1-mG_2a_ against CHO/CD44v3–10, MKN45, and COLO205 xenografts were evaluated. Following the inoculation of CHO/CD44v3–10, C_44_Mab-1-mG_2a_ or control mIgG_2a_ was intraperitoneally administered into CHO/CD44v3–10 xenograft-bearing mice on days 7, 14, and 21. The tumor volume was measured on days 7, 10, 14, 17, 21, 23, and 28 after the inoculation. The C_44_Mab-1-mG_2a_ administration resulted in a potent and significant reduction in CHO/CD44v3–10 xenografts on days 23 (*p* < 0.05) and 28 (*p* < 0.01) compared with that of mIgG_2a_ (Figure 7A). In the cases of MKN45 and COLO205 xenografts, C_44_Mab-1-mG_2a_ or control mIgG_2a_ was intraperitoneally administered on days 7 and 14. The significant reductions were observed in MKN45 xenografts on days 10 (*p* < 0.05), 14 (*p* < 0.01), and 21 (*p* < 0.01) compared with that of mIgG_2a_ (Figure 7B). The significant reductions were also observed in COLO205 xenografts on days 14 (*p* < 0.05) and 21 (*p* < 0.01) compared with that of mIgG_2a_ (Figure 7C).

Significant decreases in xenograft weight caused by C_44_Mab-1-mG_2a_ were observed in CHO/CD44v3–10 xenografts (52% reduction; *p* < 0.05; Figure 7D), MKN45 xenografts (24% reduction; *p* < 0.05; Figure 7E), and COLO205 xenografts (23% reduction; *p* < 0.05; Figure 7F). Body weight loss was not observed in the xenograft-bearing mice treated with C_44_Mab-1-mG_2a_ (Figure 7G–I).

## 3. Discussion

Given the critical role of CD44 in cancer metastasis and therapeutic resistance, several strategies targeting CD44 have been developed for the treatment of diverse malignancies, including head and neck, breast, gynecological, and ovarian cancers [25]. However, clinical trials evaluating the safety and efficacy of these interventions have demonstrated limited success. RG7356, an anti-pan CD44 mAb, demonstrated an acceptable safety profile. However, the trial was discontinued due to the absence of a clinically significant or dose-dependent response [26]. Clinical evaluation of an antibody–drug conjugate (ADC) comprising an anti-CD44v6 mAb (bivatuzumab–mertansine) was performed, but development was halted because of severe cutaneous toxicity [27,28]. In this study, a mouse IgG_2a_ version of anti-CD44v9 mAb, C_44_Mab-1-mG_2a,_ was developed (Figure 1). C_44_Mab-1-mG_2a_ showed a high reactivity to CD44v9-positive gastric and colorectal cancer cell lines (Figure 3) and exhibited the in vitro (Figure 5 and Figure 6) and in vivo antitumor activities (Figure 7) in their xenograft models. Although the potent antitumor effect was observed in CHO/CD44v3–10 tumors, the effects against MKN45 and COLO205 tumors were moderate. As shown in Figure 5 and Figure 6, the ADCC and CDC against CHO/CD44v3–10 were potently elicited compared to MKN45 and COLO205. Therefore, these differences might reflect the antitumor efficacy. These findings underscore the need for a more comprehensive understanding of CD44 biology to enhance therapeutic efficacy while minimizing side effects.

In a gastric cancer cell line, the predominant CD44v transcripts—CD44v3, 8–10, CD44v6–10, CD44v8–10, and CD44v3, 8—were identified [29]. C_44_Mab-1-mG_2a_ would recognize almost all products derived from these transcripts, enabling detection of a broad spectrum of CD44v-expressing gastric cancer cells. Among these variants, CD44v8–10 has been shown to play a critical role in regulating ROS defense and in promoting gastric cancer progression [12]. The CD44v8–10 stabilizes xCT, an essential component of the cellular antioxidant defense system, thereby allowing cancer stem cells to mitigate oxidative stress, confer ferroptosis resistance, and maintain their tumorigenic capacity [30]. Accordingly, an anti-CD44v9 mAb, clone RV3, has predominantly been used for immunohistochemistry [12]. The staining pattern was similar to that of C_44_Mab-1-mG_2a_ (Figure 4). Previous studies have demonstrated that CD44v9 serves as a predictive marker for gastric cancer recurrence [31] and as a biomarker for patient selection and therapeutic efficacy of the xCT inhibitor sulfasalazine [32]. Further studies using C_44_Mab-1-mG_2a_ are warranted to clarify the association between CD44v9 expression and clinicopathological features. In addition, C_44_Mab-1-mG_2a_ detected both intestinal-type gastric cancer and diffuse-type gastric cancer in immunohistochemistry (Figure 4). Future investigations should determine whether CD44v9 expression is enriched in specific molecular subtypes of gastric cancer [33]. In gastric cancer with peritoneal metastasis, CD44 was identified as an amplified cancer driver gene [10]. It is worthwhile to evaluate the effect of C_44_Mab-1-mG_2a_ in a peritoneal metastasis model of gastric cancer.

CD44 is reported to be a target gene of Wnt/β-catenin in a mouse intestinal tumor model [34]. However, the mechanism of the v9 inclusion during colorectal cancer development remains to be determined. Large-scale genomic analyses have revealed that colorectal cancers were classified into four subtypes: canonical, mesenchymal, metabolic, and microsatellite instability immune types [35]. The relationship between CD44v9 and the subtypes should be investigated in the future. We previously examined the CD44v9 expression on colorectal cancer tissues by immunohistochemistry and found that the CD44v9 was expressed on the basolateral surface of colorectal cancers [22]. The basolateral CD44 expression was previously reported and co-localized with HA [36], Annexin II [37], and Claudin-7-EpCAM complex [38]. Therefore, the basolateral expression of CD44 may serve as an adhesion-mediated signal transduction, which contributes to colorectal cancer tumorigenesis.

Since bivatuzumab–mertansine showed toxicity to normal skin epithelium [27,28], CD44v6-targeted strategies have been further developed to chimeric antigen receptor T (CAR-T) cell therapy. CD44v6 CAR-T cells exhibited antitumor activity against primary human multiple myeloma and acute myeloid leukemia [39]. Moreover, CD44v6 CAR-T cells effectively suppressed xenograft tumor growth in models of lung and ovarian carcinomas [40], highlighting their potential for application in solid tumor therapy. Although CD44v9 expression is minimally detected in normal colonic epithelium by C_44_Mab-1, it is observed in other normal tissues, such as squamous epithelium [22]. Therefore, to enable the therapeutic application of C_44_Mab-1, additional studies are required to mitigate potential toxicities in these tissues.

To reduce the reactivity and toxicities in normal tissues, cancer-specific mAbs (CasMabs) for various antigens have been developed. More than three hundred anti-human epidermal growth factor receptor 2 (HER2) mAb clones have been established by immunization of mice with cancer cell-expressed HER2. Among them, H_2_CasMab-2 (H_2_Mab-250) was screened for cancer-specific reactivity using flow cytometry. H_2_CasMab-2 specifically recognized HER2 expressed on breast cancer cells but not on normal epithelial cells from lung bronchus, kidney proximal tubule, colon, and mammary gland. Epitope analysis revealed the mode of cancer-specific recognition [41]. Furthermore, chimeric antigen receptor (CAR)-T cell therapy using a single chain variable fragment of H_2_CasMab-2 has been developed and evaluated in a phase I clinical trial (NCT06241456) for patients with HER2-positive advanced solid tumors [42]. Although we additionally obtained several clones of anti-CD44v9 mAbs, further clones are required for the CD44v9 CasMab screening. C_44_Mab-1-mG_2a_ would serve as a reference mAb when the CD44v9 CasMabs are established in the future.

## 4. Materials and Methods

### 4.1. Cell Lines

CHO-K1 and COLO205 (colorectal cancer) cell lines were obtained from the American Type Culture Collection (Manassas, VA, USA). MKN45 and NUGC-4 (human gastric cancer) cell lines were obtained from the Japanese Collection of Research Bioresources (Osaka, Japan). KatoIII (human gastric cancer) was obtained from the Cell Resource Center for Biomedical Research Institute of Development, Aging, and Cancer at Tohoku University (Miyagi, Japan). CHO/CD44v3–10 was established previously [43].

MKN45 and NUGC-4 were maintained in Dulbecco’s Modified Eagle’s Medium (DMEM; Nacalai Tesque, Inc., Kyoto, Japan). CHO/CD44v3–10, CHO-K1, KatoIII, and COLO205 were cultured in Roswell Park Memorial Institute (RPMI) 1640 medium (Nacalai Tesque, Inc.). All culture media were supplemented with 10% heat-inactivated fetal bovine serum (FBS; Thermo Fisher Scientific Inc., Waltham, MA, USA), 100 U/mL penicillin, 100 μg/mL streptomycin, and 0.25 μg/mL amphotericin B (Nacalai Tesque, Inc.). Cells were incubated at 37 °C in a humidified atmosphere containing 5% CO_2_ and 95% air.

### 4.2. Recombinant mAb Production

A control mIgG_2a_ mAb, PMab-231 (mouse IgG_2a_, κ, an anti-tiger podoplanin mAb) was previously produced [23]. An anti-CD44v9 mAb, C_44_Mab-1 (mouse IgG_1_, κ) was established previously [22]. To create the mouse IgG_2a_ version (C_44_Mab-1-mG_2a_), the V_H_ cDNA of C_44_Mab-1 and the C_H_ of mouse IgG_2a_ were cloned into the pCAG-Neo vector [FUJIFILM Wako Pure Chemical Corporation (Wako), Osaka, Japan]. Similarly, the V_L_ cDNA of C_44_Mab-1 and the C_L_ of the mouse kappa chain were cloned into the pCAG-Ble vector (Wako). The antibody expression vectors were transfected into BINDS-09 (fucosyltransferase 8-knockout ExpiCHO-S) using the ExpiCHO Expression System (Thermo Fisher Scientific Inc.) to produce C_44_Mab-1-mG_2a_ and PMab-231. All antibodies were purified using Ab-Capcher (ProteNova Co., Ltd., Kagawa, Japan).

### 4.3. Flow Cytometry

Cells were detached by brief incubation with a solution containing 1 mM ethylenediaminetetraacetic acid (EDTA; Nacalai Tesque, Inc.) and 0.25% trypsin. Following centrifugation and washing with phosphate-buffered saline (PBS) containing 0.1% bovine serum albumin (BSA) as a blocking buffer, the cells were incubated with primary mAbs for 30 min at 4 °C. Subsequently, cells were treated with Alexa Fluor 488-conjugated anti-mouse IgG secondary antibody (1:2000; Cell Signaling Technology, Inc., Danvers, MA, USA). Fluorescence signals were acquired using an SA3800 Cell Analyzer (Sony Corp., Tokyo, Japan). The dissociation constant *(K*_D_) value was determined by flow cytometry as described previously [22].

### 4.4. Immunohistochemical Analysis

Formalin-fixed paraffin-embedded tissue arrays of gastric cancer (BS01011b) were purchased from US Biomax Inc. (Rockville, MD, USA). The tissue arrays were autoclaved in citrate buffer (pH 6.0; Nichirei Biosciences, Inc., Tokyo, Japan) for 20 min. The blocking was performed using SuperBlock T20 (Thermo Fisher Scientific, Inc.). The sections were incubated with 5 μg/mL of C_44_Mab-1-mG_2a_ and 5 μg/mL of an anti-pan-CD44 mAb, C_44_Mab-46-mG_2a_ [44]. Then, the tissue arrays were treated with the EnVision+ Kit for mouse (Agilent Technologies Inc., Santa Clara, CA, USA). The chromogenic reaction was performed using 3,3′-diaminobenzidine tetrahydrochloride (DAB; Agilent Technologies Inc.). Hematoxylin (FUJIFILM Wako Pure Chemical Corporation) was used for counterstaining. A Leica DMD108 (Leica Microsystems GmbH, Wetzlar, Germany) was used to obtain images.

### 4.5. ADCC by C_44_Mab-1-mG_2a_

The animal experiments were conducted in accordance with the institutional regulations and ethical guidelines to minimize pain and distress, and were approved by the Institutional Committee for Animal Experiments of the Institute of Microbial Chemistry (Numazu, Japan; approval number: 2025-029). Five-week-old female BALB/c nude mice were purchased from Japan SLC Inc. (Shizuoka, Japan). The splenocytes (designated as effector cells) were prepared as described previously [45]. The ADCC activity of C_44_Mab-1-mG_2a_ was investigated as follows. Calcein AM-labeled target cells (CHO/CD44v3–10, MKN45, and COLO205) were co-incubated with the effector cells at an effector-to-target (E:T) ratio of 50:1 in the presence of 100 μg/mL of C_44_Mab-1-mG_2a_ (*n* = 3) or control mIgG_2a_ (*n* = 3). Following a 4.5 h incubation, the Calcein release into the medium was measured using a microplate reader (Power Scan HT; BioTek Instruments, Inc., Winooski, VT, USA).

Cytotoxicity was calculated as a percentage of lysis using the following formula: % lysis = (E − S)/(M − S) × 100, where E represents the fluorescence intensity from co-cultures of effector and target cells, S denotes the spontaneous fluorescence from target cells alone, and M corresponds to the maximum fluorescence obtained after complete lysis using a buffer containing 10 mM Tris-HCl (pH 7.4), 10 mM EDTA, and 0.5% Triton X-100. Data are presented as mean ± standard error of the mean (SEM). Statistical significance was evaluated using a two-tailed unpaired *t*-test.

### 4.6. CDC by C_44_Mab-1-mG_2a_

The Calcein AM-labeled target cells (CHO/CD44v3–10, MKN45, and COLO205) were plated and mixed with rabbit complement (final dilution 10%, Low-Tox-M Rabbit Complement; Cedarlane Laboratories, Hornby, ON, Canada) and 100 μg/mL of C_44_Mab-1-mG_2a_ (*n* = 3) or control mIgG_2a_ (*n* = 3). Following incubation for 4.5 h at 37 °C, the Calcein release into the medium was measured, as described above.

### 4.7. Antitumor Activity of C_44_Mab-1-mG_2a_

The Institutional Committee approved the animal experiment for Animal Experiments of the Institute of Microbial Chemistry (approval number: 2025-011). Throughout the study, mice were housed under specific pathogen-free conditions with an 11 h light/13-h dark cycle and provided with food and water ad libitum. Body weight was measured twice weekly, and general health status was assessed three times per week. Humane endpoints were predefined as a body weight loss exceeding 25% of the initial weight and/or a tumor volume exceeding 3000 mm^3^.

Female BALB/c nude mice (4 weeks old) were obtained from Japan SLC, Inc. Tumor cells (0.3 mL of a 1.33 × 10^8^ cells/mL suspension in DMEM) were mixed with 0.5 mL of BD Matrigel Matrix Growth Factor Reduced (BD Biosciences, San Jose, CA, USA). A 100 μL aliquot of the mixture, containing 5 × 10^6^ cells, was subcutaneously injected into the left flank of each mouse (day 0). To evaluate the antitumor activity of C_44_Mab-1-mG_2a_ against CHO/CD44v3–10, 100 μg of C_44_Mab-1-mG_2a_ (*n* = 8) or control mIgG_2a_ (*n* = 8) diluted in 100 μL of PBS was administered intraperitoneally to tumor-bearing mice on day 7 post-inoculation. The mAbs were further administered on days 14 and 21. Mice were euthanized on day 28 following tumor cell implantation. To evaluate the antitumor activity of C_44_Mab-1-mG_2a_ against MKN45 and COLO205, 100 μg of C_44_Mab-1-mG_2a_ (*n* = 8) or control mIgG_2a_ (*n* = 8) diluted in 100 μL of PBS was administered intraperitoneally to tumor-bearing mice on day 7 post-inoculation. A second dose was administered on day 14. Mice were euthanized on day 21 following tumor cell implantation.

Tumor size was measured, and volume was calculated using the formula: volume = W^2^ × L/2, where W represents the short diameter and L the long diameter. Data are presented as the mean ± standard error of the mean (SEM). Statistical analysis was performed using one-way ANOVA followed by Sidak’s post hoc test. A *p*-value < 0.05 was considered statistically significant.

## Figures and Tables

**Figure 1 ijms-26-09170-f001:**
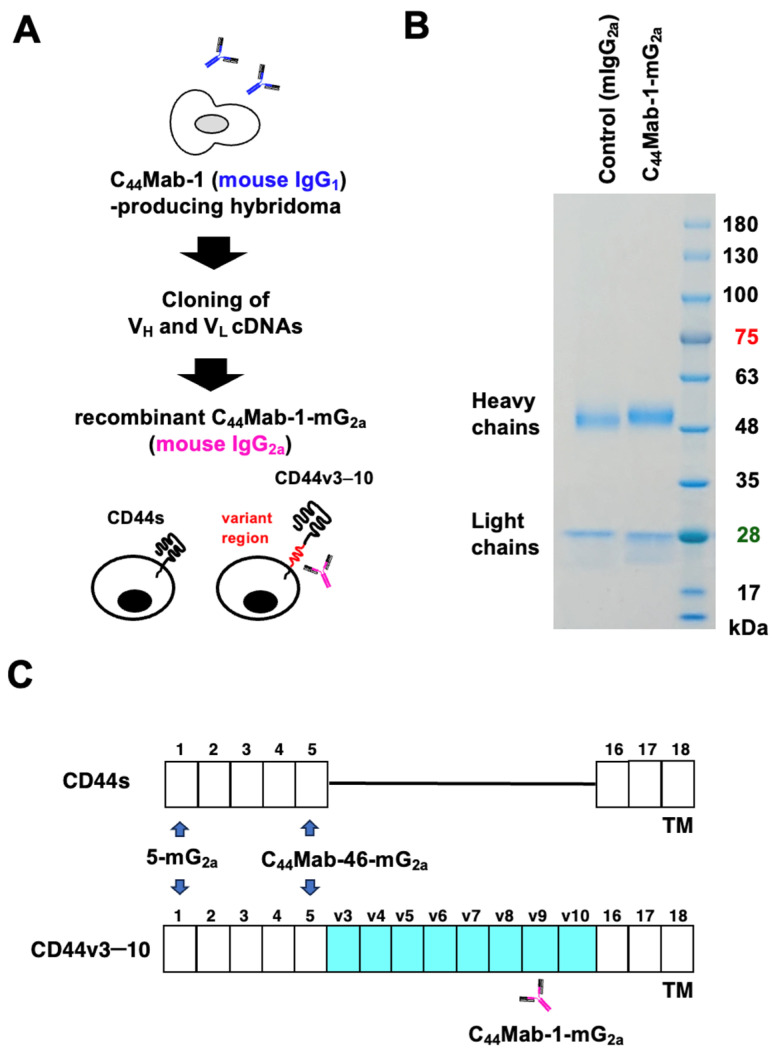
Production of a recombinant anti-CD44v9 mAb, C_44_Mab-1-mG_2a,_ from C_44_Mab-1-producing hybridoma. (**A**) After determination of CDRs of C_44_Mab-1 (mouse IgG_1_), recombinant C_44_Mab-1-mG_2a_ (mouse IgG_2a_) was produced and purified. (**B**) C_44_Mab-1-mG_2a_ and PMab-231 (control mIgG_2a_) were treated with sodium dodecyl sulfate sample buffer containing 2-mercaptoethanol. Proteins were separated on a polyacrylamide gel. The gel was stained with Bio-Safe CBB G-250 Stain. (**C**) Extracellular structure of CD44s and CD44v3–10. Standard exons (1–5 and 16–18) and variant exons (v3-v10)-encoded regions are presented. C_44_Mab-1-mG_2a_ recognizes v9-encoded region. Pan-CD44 mAbs (5-mG_2a_ and C_44_Mab-46-mG_2a_) recognize both CD44s and CD44v including CD44v3–10. TM, transmembrane.

**Figure 2 ijms-26-09170-f002:**
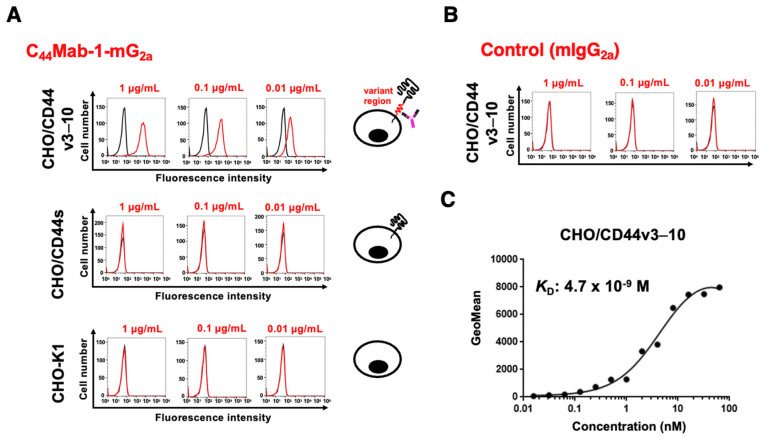
Flow cytometry analysis using C_44_Mab-1-mG_2a_. (**A**) CHO/CD44v3–10, CHO/CD44s, and CHO-K1 were treated with 0.01, 0.1, and 1 µg/mL of C_44_Mab-1-mG_2a_. (**B**) CHO/CD44v3–10 was treated with 0.01, 0.1, and 1 µg/mL of control mIgG_2a_. Then, the cells were treated with Alexa Fluor 488-conjugated anti-mouse IgG. Fluorescence data were analyzed using the SA3800 Cell Analyzer. (**C**) CHO/CD44v3–10 were treated with serially diluted C_44_Mab-1-mG_2a_, followed by Alexa Fluor 488-conjugated anti-mouse IgG treatment. The fluorescence data were analyzed, and the *K*_D_ values were determined.

**Figure 3 ijms-26-09170-f003:**
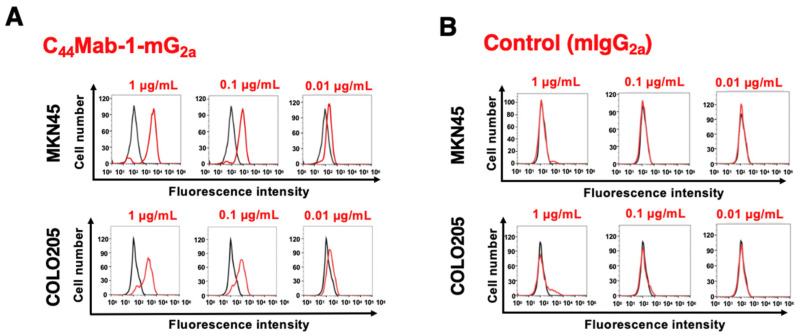
Flow cytometry analysis of C_44_Mab-1-mG_2a_ to CD44v9-positive cancer cell lines. (**A**) MKN45 and COLO205 were treated with 0.01, 0.1, and 1 µg/mL of C_44_Mab-1-mG_2a_. (**B**) MKN45 and COLO205 were treated with 0.01, 0.1, and 1 µg/mL of control mIgG_2a_. Then, the cells were treated with Alexa Fluor 488-conjugated anti-mouse IgG. Fluorescence data were analyzed using the SA3800 Cell Analyzer.

**Figure 4 ijms-26-09170-f004:**
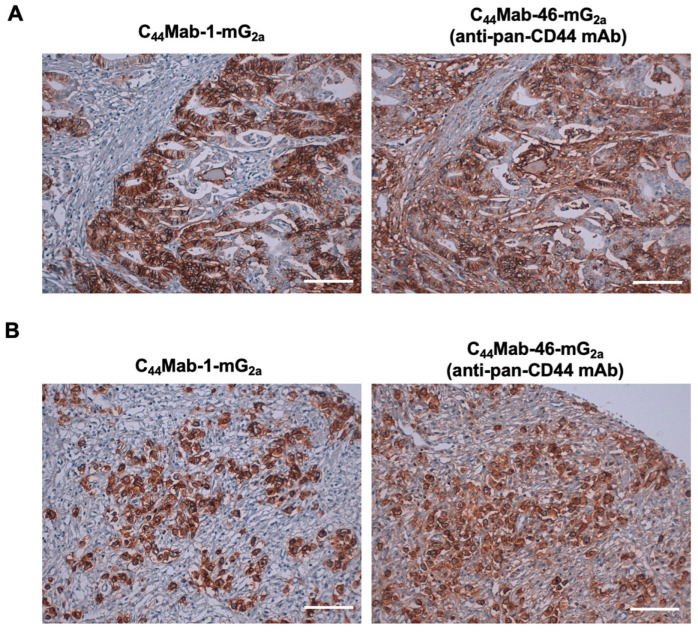
Immunohistochemistry of gastric cancer using C_44_Mab-1-mG_2a_. Immunohistochemical analysis using C_44_Mab-1-mG_2a_ and C_44_Mab-46-mG_2a_ (a pan-CD44 mAb) against gastric cancer tissue array (BS01011b). After antigen retrieval, serial sections of the tissue arrays were incubated with 5 µg/mL of C_44_Mab-1-mG_2a_ or 5 µg/mL of C_44_Mab-46-mG_2a,_ followed by treatment with the Envision+ kit. The color was developed using 3,3′-diaminobenzidine tetrahydrochloride (DAB), and the sections were counterstained with hematoxylin. Scale bar = 100 µm. Typical staining patterns of intestinal-type gastric cancer ((**A**), no. 8) and diffuse-type gastric cancer ((**B**), no. 58) were shown.

**Figure 5 ijms-26-09170-f005:**
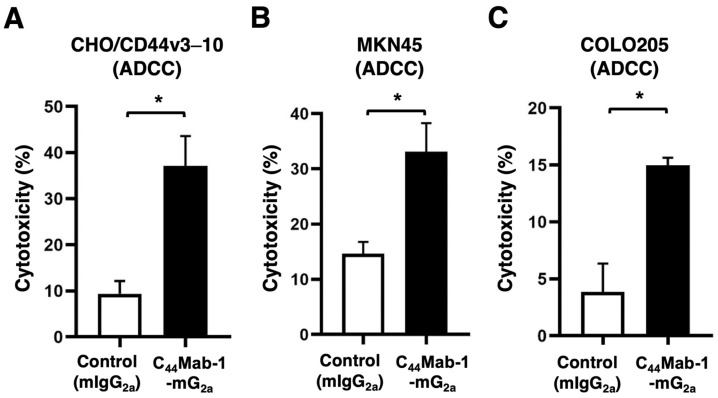
ADCC by C_44_Mab-1-mG_2a_ against CHO/CD44v3–10, MKN45, and COLO205. ADCC induced by C_44_Mab-1-mG_2a_ or control mouse IgG_2a_ (mIgG_2a_) against CHO/CD44v3–10 (**A**), MKN45 (**B**), and COLO205 (**C**). Calcein AM-labeled target cells (CHO/CD44v3–10, MKN45, and COLO205) were incubated with the effector splenocytes in the presence of 100 μg/mL of C_44_Mab-1-mG_2a_ or control mIgG_2a_. Following a 4.5 h incubation, the Calcein release into the medium was measured. Values are shown as mean ± SEM. Asterisks indicate statistical significance (* *p* < 0.05; Two-tailed unpaired *t*-test).

**Figure 6 ijms-26-09170-f006:**
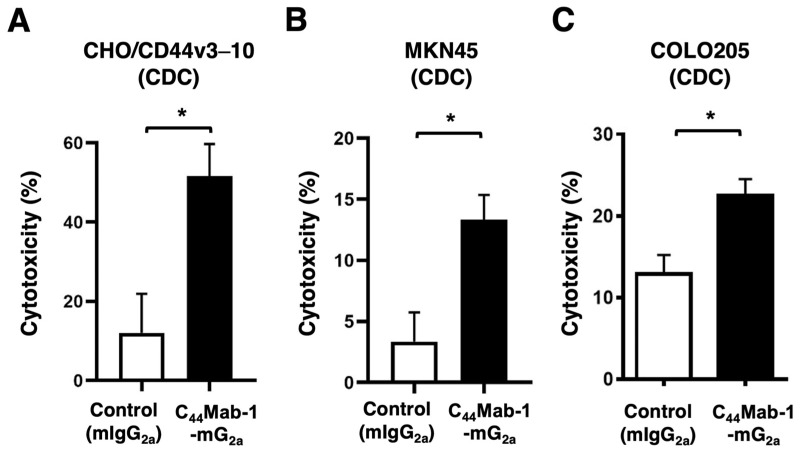
CDC by C_44_Mab-1-mG_2a_ against CHO/CD44v3–10, MKN45, and COLO205. CDC induced by C_44_Mab-1-mG_2a_ or control mouse IgG_2a_ (mIgG_2a_) against CHO/CD44v3–10 (**A**), MKN45 (**B**), and COLO205 (**C**). Calcein AM-labeled target cells (CHO/CD44v3–10, MKN45, and COLO205) were incubated with complements and C_44_Mab-1-mG_2a_ or control mIgG_2a_. Following incubation for 4.5 h at 37 °C, the Calcein release into the medium was measured. Values are shown as mean ± SEM. Asterisks indicate statistical significance (* *p* < 0.05; Two-tailed unpaired *t*-test).

**Figure 7 ijms-26-09170-f007:**
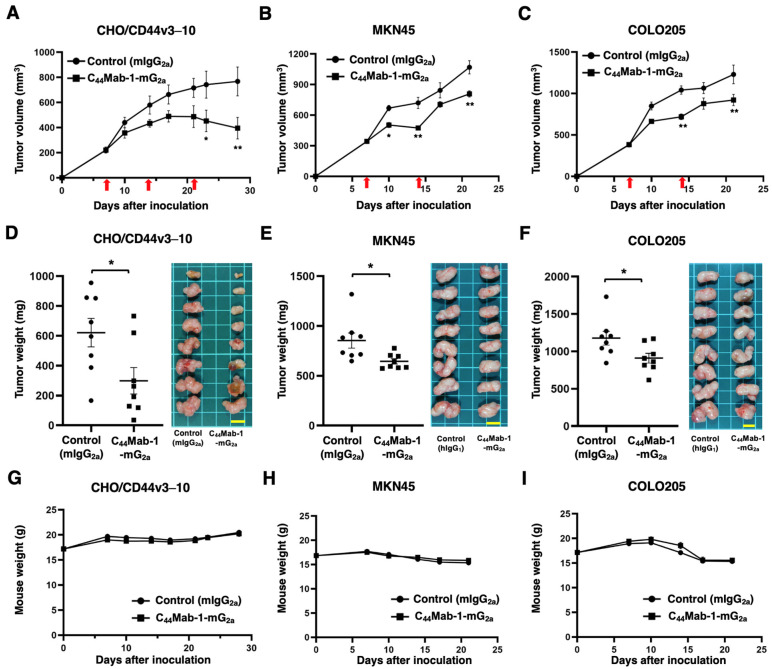
Antitumor activity of C_44_Mab-1-mG_2a_ against CHO/CD44v3–10, MKN45, and COLO205 xenograft. (**A**–**C**) CHO/CD44v3–10 (**A**), MKN45 (**B**), and COLO205 (**C**) were subcutaneously injected into BALB/c nude mice (day 0). An amount of 100 μg of C_44_Mab-1-mG_2a_ or control mouse IgG_2a_ (mIgG_2a_) was intraperitoneally injected into each mouse on day 8. Additional antibodies were injected on day 14 (MKN45 and COLO205) or 14 and 21 (CHO/CD44v3–10) (red arrows). The tumor volume is represented as the mean ± SEM. ** *p* < 0.01, * *p* < 0.05 (ANOVA with Sidak’s multiple comparisons test). (**D**–**F**) The mice treated with the mAbs were euthanized on day 21 (MKN45 and COLO205) or 28 (CHO/CD44v3–10). The xenograft weights were measured. Values are presented as the mean ± SEM. ** *p* < 0.01 (Two-tailed unpaired *t*-test). (**G**–**I**) Body weights of xenograft-bearing mice treated with the mAbs. There is no statistical difference.

## Data Availability

The data presented in this study are available in the article and Appendix A.

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
