# Peer review of "Antitumor Activity by an Anti-CD44 Variant 9 Monoclonal Antibody in Gastric and Colorectal Cancer Xenograft Models"

_ijms, 2025, doi:10.3390/ijms26189170_

Round 1
Reviewer 1 Report
Comments and Suggestions for Authors
The manuscript by Tawara et al describes the anti-tumour activity of an antibody targeted to the variant 9 region of CD44, with this region appearing in a number of CD44 variants. The antibody was isolated in a previously published work, which defined the epitope as the variant 9 region and showed that it binds to both recombinant CD44v3-10 (expressed in CHO cells) and native CD44 in COLO201 and COLO205 cells. This publication expands on the characterisation by reformatting the antibody to a mouse IgG2a (from a mouse IgG1) to give the antibody effector function activity, then testing the antibody’s ability to induce ADCC and CDC in cells expressing recombinant or native CD44 with the variant 9 region. They also tested whether treatment with the antibody reduced tumour growth in xenograft mouse models.
Please note that I was unable to view the supplementary materials – there was no separate attachment and the link in the PDF did not work. I have accepted the authors’ description of the results without viewing the figures.
The study is well described and the results are clearly demonstrated. The work is significant since splice variants of CD44 have been shown to be involved in tumour metastasis, and antibody treatment would need to be specific for splice variants rather than the standard from of CD44 (CD44s) to avoid targeting normal cells.
Suggested improvements and comments:
A diagram of the CD44 structure showing how the different variants (CD44s, CD44v3-10, CD44v8-10 etc) relate to each other would be helpful, especially since it is not immediately obvious that the variant 9 region refers to a particular exon which is present in several different CD44 variants.
Lines 95-96: This sentence is ambiguous. It would be better changed to: ‘Previously we isolated an anti-CD44v9 mAb, C44Mab-1, by immunizing mice with CHO/CD44v3-10’.
Line 119-120: It is stated that the mG2a version has comparable reactivity and affinity with the parental IgG1 mAb. However there is no report of the IgG1 affinity in this paper. Are you referring to the affinity reported in previous work? If so please cite the reference. Or show the data if the IgG1 was measured in the same experiment.
Figure 2C: What is GeoMean on the y-axis? Should this be Mean Fluorescence Intensity?
Paragraph, Lines 129-135: Are these cell lines (COLO205, MKN45, LMSU, KatoIII, NUGC-4) known to be CD44v9 positive?
Figure 5 and Figure 6: It would be good to include CHO cells alone (or another cell line not expressing CD44v9) in the ADCC and CDC assays as a negative control to verify that the results are dependent on the presence of CD44v9.
Figure 7A, B and C: It would be good to indicate on the graphs the days where antibody was administered.
Line 239: Which gastric cancer cell line?
Line240-241: It is stated that C44Mab-1-mG2a recognises almost all products derived from CD44v3,8-10, CD44v6-10, CD44v8-10 and CD44v3,8, but the results in this paper only specifically report data for CD44v8-10. Please clarify if this is referring to results from this study, or a previous study (although the previous study used IgG1). Or are you inferring that it would bind to all of these except CD44v3,8 due to the presence of v9?
Line 247: Is there a reference for mAb RV3? How is this mab different in behaviour to C44Mab-1?
Line 301: Please include a short description of CHO/CD44v3-10. Is it a stable cell line expressing recombinant CD44v3-10?
Line 317-318: Please add here, or mention in the discussion, why this cell line was used? Was it to increase the ADCC activity? Where was this cell line obtained from?
Reviewer 2 Report
Comments and Suggestions for Authors
In general, a well-planned and methodologically advanced study has been submitted for review. However, there are a number of comments that need to be addressed, namely:
- It is unclear what the CHO/CD44v3-10 cell line used in the work is. Is it a normal or a tumor cell line? It is necessary to describe in more detail in this study the production of this line in "Materials and Methods".
- According to Figure 7, the antitumor effect of the new antibody was mainly manifested in the inhibition of intraperitoneal growth of the CHO/CD44v3-10 cell line, and the antitumor effect of the antibody on other cancer cell lines was expressed very moderately. This needs to be discussed in the "Discussion" chapter.
- The block of studies on the possibility of using this new antibody for IHC diagnostics of human gastric cancer raises questions. It is impossible to see Additional Table 1 in the submitted version of the article. How good is it that only in 72% of cases did the new antibodies detect tumor parenchyma? In my opinion, the discussion should present comparative data on IHC diagnostics of gastric cancer with previously developed antibodies.
Conclusion, minor revision.
